# A New Class of Weighted CUSUM Statistics

**DOI:** 10.3390/e24111652

**Published:** 2022-11-14

**Authors:** Xiaoping Shi, Xiang-Sheng Wang, Nancy Reid

**Affiliations:** 1Department of Computer Science, Mathematics, Physics and Statistics, University of British Columbia, Kelowna, BC V1V 1V7, Canada; 2Department of Mathematics, University of Louisiana at Lafayette, Lafayette, LA 70503, USA; 3Department of Statistical Sciences, University of Toronto, Toronto, ON M5S 3G3, Canada

**Keywords:** asymptotic distribution, Brownian bridge, exact distribution, quadratic weights, weak dependence

## Abstract

A change point is a location or time at which observations or data obey two different models: before and after. In real problems, we may know some prior information about the location of the change point, say at the right or left tail of the sequence. How does one incorporate the prior information into the current cumulative sum (CUSUM) statistics? We propose a new class of weighted CUSUM statistics with three different types of quadratic weights accounting for different prior positions of the change points. One interpretation of the weights is the mean duration in a random walk. Under the normal model with known variance, the exact distributions of these statistics are explicitly expressed in terms of eigenvalues. Theoretical results about the explicit difference of the distributions are valuable. The expansions of asymptotic distributions are compared with the expansion of the limit distributions of the Cramér-von Mises statistic and the Anderson and Darling statistic. We provide some extensions from independent normal responses to more interesting models, such as graphical models, the mixture of normals, Poisson, and weakly dependent models. Simulations suggest that the proposed test statistics have better power than the graph-based statistics. We illustrate their application to a detection problem with video data.

## 1. Introduction

A change point is a location or time at which observations or data obey two different models: before and after. Detecting change points is a nontrivial problem and has been studied by many authors; see a book treatment in [1] and recent advances in CUSUM-based change point tests [2,3,4]. In real problems, we may know some prior information about the location of the change point, say at the right or left tail of the sequence. How does one incorporate prior information into current CUSUM-based statistics? We consider a new class of weighted CUSUM statistics for a simple model and provide some extensions to more complicated models.

Given a series of univariate random variables $Y_{1},\ldots ,Y_{n}$, we consider the problem of testing whether there is a change in the mean of their distribution. The test statistic we use is:(1)$$S_{n}\left(Y;\tau ,\gamma \right)=\sum_{k=1}^{n-1}w_{k}^{-1}\left(\tau \right){\left\{\sum_{i=1}^{k}\left(Y_{i}-\bar{Y}\right)\right\}}^{\gamma },$$
where $Y={\left(Y_{1},\ldots ,Y_{n}\right)}^{⊤}$, $\bar{Y}=n^{-1}\sum_{j=1}^{n}Y_{j}$, γ>0, and
(2)$$w_{k}\left(\tau \right)=-{\left(k-\tau \right)}^{2}+\max\left\{\tau^{2},{\left(n-\tau \right)}^{2}\right\}=\left\{\begin{array}{c} (n+k)(n-k),\;\mathrm{if}\;\tau =0,\\ k(n-k),\;\mathrm{if}\;\tau =n/2,\\ k(2n-k),\;\mathrm{if}\;\tau =n, \end{array}\right.$$
where τ=0,n/2, and *n* account for three different prior positions of the change point, respectively. We call $S_{n}$ a weighted CUSUM (WC) statistic.

Inspired by the change point literature, we consider these types of quadratic weights. The term $\max\left\{\tau^{2},{\left(n-\tau \right)}^{2}\right\}=\max_{0\leq j\leq n}{\left(j-\tau \right)}^{2}$ is introduced to ensure that the weight $w_{k}\left(\tau \right)$ is positive for any 0<k<n. Usually, we choose γ=2 to capture the change in the mean. When τ=n/2, the weight $w_{k}\left(n/2\right)=k\left(n-k\right)$ corresponds to the likelihood ratio test; see Csörgö and Horváth [1] and a related review in Jandhyala et al. [5]. If prior information indicates that the change point more likely occurs in the right or left tail of the sequence, we can set the weight $w_{k}\left(0\right)=\left(n+k\right)\left(n-k\right)$ (left drifted to the symmetry center point 0) or $w_{k}\left(n\right)=k\left(2n-k\right)$ (right drifted to the symmetry center point *n*) to improve the power of the test.

One interpretation of the weights is the mean duration in a random walk $\left\{X_{i},i\geq 0\right\}$ on N+1 states, {0,1,…,N}, whose transition probability is given by $P(X_{i+1}=k\pm 1|X_{i}=k)=1/2$ for k=1,…,N−1, $P(X_{i+1}=0|X_{i}=0)=1,$ and $P(X_{i+1}=N|X_{i}=N)=1$. Let *T* denote the random time at which the process first reaches 0 or *N*. Then, for k=1,…,n−1, $E\left(T|X_{0}=k\right)=k\left(n-k\right)=w_{k}\left(n/2\right)$ if N=n; $E\left(T|X_{0}=k\right)=k\left(2n-k\right)=w_{k}\left(n\right)$ if N=2n; and $E\left(T|X_{0}=n-k\right)=\left(n+k\right)\left(n-k\right)=w_{k}\left(0\right)$ if N=2n. Figure 1 depicts four vectors $w_{k}$ for n=10. The centers of symmetry of these quadratic weights are at different positions.

The weights in (Equation 1) can be thought of as an inverse prior probability on the change point, giving $S_{n}$ a Bayesian flavor, as in Gardner [6], who used the uniform prior $n^{-2}$, or Perron [7], who devised a unit-root test for time series. From a frequentist perspective, the weighted sum statistic offers an alternative to the maximum statistic most commonly used Csörgö and Horváth [1], which we show (in small simulations omitted here) has higher power, especially when the change point is at the center of the sequence for any τ, in the right tail of the sequence for τ=n, and in the left tail of the sequence for τ=0.

For these types of quadratic weights, a couple of questions naturally arise: will different weights lead to different distributions of WC in Equation (Equation 1)? If so, how significant will the differences in the distribution be? If two different weights lead to the same distribution, are there any intrinsic reasons? Although one can estimate the distribution of WC by simulation, theoretical results about the explicit differences of the distributions are valuable. Moreover, simulations and computations of eigenvalues for large *n* are computationally expensive. To answer the aforementioned questions, we shall study the distribution of the WC theoretically; we derive Karhunen–Loève expansions of the exact and asymptotic distributions of the WC statistics. The calculation of a Karhunen–Loève expansion is a nontrivial task, even under the normal model. Gardner [6] discussed the uniform weight under the normal assumption, but the quadratic weights we consider here increase the difficulty substantially. We present below a unified theory that enables us to establish the distribution of WC using dual Hahn polynomials. The asymptotic distributions for the quadratic weights $w_{k}\left(0\right)$ and $w_{k}\left(n\right)$ are identical, and the expansions of asymptotic distributions between $w_{k}\left(0\right)$ and another quadratic weight $w_{k}\left(n/2\right)$ differ by an odd number of terms. We make a comparison with the expansion of the limit distributions of the Cramér-von Mises statistic and the Anderson and Darling statistics; see also MacNeill [8].

The WC has some variants in other models. For example, in the graphical model, γ can be 1 if we replace Y with a count of edges. Here, the main challenge is to approximate the covariance of edge-count statistics under the null permutations. In the normal mixed model, a variant of WC can be derived by considering a marginal likelihood function. In the Poisson mixed model, however, the calculation of the marginal likelihood function is hindered by an integral without a closed form. To approximate this integral, one may use Laplace, or saddle point approximation [9,10,11,12,13]. Here, we apply the saddle point approximation to the integral and provide a variant of WC related to the log link. For the classical change point Poisson model without latent variables, see [1] (p. 27); for the Poisson process with a change point, we refer readers to Akman and Raftery [14], Loader [15]. Moreover, to adopt the assumption of weak dependence in practice, we avoid the estimation of the variance and provide a randomized version of WC.

The structure of the paper is outlined as follows. In Section 2, we derive the explicit expansions of the distribution of the WC statistics and explore their connections with the Karhunen–Loève expansion. We derive extended versions of WC by considering the observations as nodes in the graphical model and allowing the observations from a normal or Poisson mixed model to be weakly dependent. In Section 3, we discuss the power of the proposed WC test. In Section 4, we use simulation to compare the performance of this test with that of a graph-based test statistic. In Section 5, we present an application for video data. In Section 6, we discuss the extension to multiple change points and suggest future work on other quadratic weights.

## 2. Exact and Asymptotic Distributions of the WC Statistics

### 2.1. Explicit Distribution for a Normal Model

We assume here that $\left\{Y_{i}\right\}$ are independent following a normal distribution with a common known variance $\sigma^{2}$. The case of unknown $\sigma^{2}$ is addressed in Remark 3, and an extension relaxing the independence assumption is given in Section 2.6.

Following the derivation in Gardner [6], we write (Equation 1) as a quadratic form
(3)$$S_{n}\left(Y;\tau ,2\right)=\frac{1}{n^{2}}\sum_{k=1}^{n-1}p_{k}{\left\{\sum_{i=1}^{k}\left(n-k\right)Y_{i}-\sum_{i=k+1}^{n}kY_{i}\right\}}^{2}=\frac{1}{n^{2}}Y^{⊤}AA^{⊤}Y=Y^{⊤}QY,$$
where $p_{k}=p_{k}\left(\tau \right)=w_{k}^{-1}\left(\tau \right)$, and $n^{2}Q=AA^{⊤}$ with $A=(A_{1},\cdots ,A_{n-1})$. Here, $A_{k}=p_{k}^{1/2}{\left(n-k,\cdots ,n-k,-k,\cdots ,-k\right)}^{⊤}$ such that the first *k* entries of $A_{k}$ are $p_{k}^{1/2}\left(n-k\right)$ and the last n−k entries $-p_{k}^{1/2}k$.

By using the recurrence identity and the dual Hahn polynomial, we obtain a new exact result in terms of the eigenvalues of *Q* in (Equation 3).

**Theorem 1.** 
*Assume that $\left\{Y_{i}\right\}$ are independent normally distributed random variables with a common mean and known variance $\sigma^{2}$. The exact distribution of $S_{n}\left(Y;\tau ,2\right)$ is*

(4)
$$\frac{S_{n}\left(Y;\tau ,2\right)}{\sigma^{2}}{=}_{d}\sum_{k=1}^{n}\lambda_{k}\left(\tau \right)Z_{k}^{2},$$

*where $Z_{k}^{2}$ are independent and identically distributed normal random variables with mean zero and variance 1, $\lambda_{n}\left(\tau \right)=0$, and*

$$\lambda_{k}\left(\tau \right)=\left\{\begin{array}{c} \frac{1}{k(k+1)},\;k=1,\ldots ,n-1,\;\mathrm{if}\;\tau =n/2\\ \frac{1}{2k(2k+1)},\;k=1,\ldots ,n-1,\;\mathrm{if}\;\tau =0\;or\;n. \end{array}\right.$$



The proof of Theorem 1 is given in Appendix A. We make the following remarks.

**Remark 1.** 
*It is interesting that $\lambda_{2k}\left(n/2\right)=\lambda_{k}\left(0\right)$ for all 0<k<n/2; namely, the eigenvalues for $w_{k}\left(n/2\right)$ with even indices coincide with the eigenvalues for $w_{k}\left(0\right)$ with indices less than n/2. As the sample size increases from n to n+1, the n−1 nonzero eigenvalues are retained and the added nonzero eigenvalue must be 1/{n(n+1)} for $w_{k}\left(n/2\right)$ or 1/{2n(2n+1)} for $w_{k}\left(0\right)$ or $w_{k}\left(n\right)$. This interesting phenomenon has not been seen in the uniform weights of Gardner [6]. As far as we know, this recursive property of the eigenvalues for the non-uniform weights is new. Figure 2 depicts the pattern of eigenvalues (cross products of rows and columns) illustrated by dots for three weights $w_{k}\left(n/2\right)$ (blue), $w_{k}\left(0\right)$ (green), and $w_{k}\left(n\right)$ (purple) with the increase of n.*


**Remark 2.** 
*The distribution in (Equation 4) can be calculated numerically using Imhof’s method [16] or simulated by a Monte Carlo method, but accurate analytical approximations are potentially faster and more stable. A saddle point approximation to the distribution of quadratic forms in normal variates was studied in Kuonen [17], building on Daniels [9,18] and Lugannani and Rice [19].*


**Remark 3.** 
*When the variance $\sigma^{2}$ is unknown, we can replace $\sigma^{2}$ with a consistent estimator*

$${\hat{\sigma }}^{2}={\left(n-1\right)}^{-1}\sum_{i=1}^{n}{\left(Y_{i}-\bar{Y}\right)}^{2},$$

*by using Slutsky’s lemma. This also holds in Corollary 1. For dependent data, one issue is to give a valid estimate of the variance; see Section 2.6.*


### 2.2. Karhunen–Loève Expansion

The squared integral of a Brownian bridge arises in the study of tests for goodness-of-fit. Given a sample of independent and identically distributed random variables with an empirical distribution function $F_{n}\left(x\right)$, the statistic
$$\omega_{n}^{2}\left(\psi \right)=n\int_{-\infty }^{\infty }{\left\{F_{n}\left(t\right)-F\left(t\right)\right\}}^{2}\psi \left\{F\left(t\right)\right\}dF\left(t\right)$$
provides a test of the null hypothesis that the observations come from the distribution F(·). The Cramér-von Mises statistic has ψ(t)≡1, and the Anderson-Darling statistic has ψ(t)=1/{t(1−t)}. Here, we shall discuss two new weights: ψ(t)=1/{t(2−t)} and $\psi \left(t\right)=1/\left(1-t^{2}\right)$.

MacNeill [8] showed that
$$\int_{0}^{1}{\left\{B\left(t\right)-tB\left(1\right)\right\}}^{2}dt=\sum_{k=1}^{\infty }\frac{1}{k^{2}\pi^{2}}Z_{k}^{2},$$
using a Fourier expansion of $B\left(t\right)-tB\left(1\right)=\sum_{k=1}^{\infty }\sqrt{2}\sin\left(k\pi t\right)/\left(k\pi \right)Z_{k}$, where $\{\sqrt{2}\sin\left(k\pi t\right),$k=1,2,⋯,∞} is an orthonormal basis in $L^{2}\left(0,1\right)$ and B(t) is a standard Brownian motion and B(t)−tB(1) is a Brownian bridge.

Anderson and Darling [20] showed that
$$\int_{0}^{1}\frac{{\left\{B\left(t\right)-tB\left(1\right)\right\}}^{2}}{t(1-t)}dt=\sum_{k=1}^{\infty }\frac{1}{k(k+1)}Z_{k}^{2}.$$

In Appendix B, we use Jacobi polynomials to derive the Karhunen–Loève expansion for the integrals of the weighted square of the Brownian bridge with two new weights ψ(t)=1/{t(2−t)} and $\psi \left(t\right)=1/\left(1-t^{2}\right)$. The results are stated in the following theorem.

**Theorem 2.** 
*The two weights ψ(t)=1/{t(2−t)} and $\psi \left(t\right)=1/\left(1-t^{2}\right)$ lead to the same Karhunen–Loève expansions:*

(5)
$$\begin{array}{c} \int_{0}^{1}\frac{{\left\{B\left(t\right)-tB\left(1\right)\right\}}^{2}}{2t-t^{2}}dt=\sum_{k=1}^{\infty }\frac{1}{2k(2k+1)}Z_{k}^{2} \end{array}$$

*and*

(6)
$$\begin{array}{c} \int_{0}^{1}\frac{{\left\{B\left(t\right)-tB\left(1\right)\right\}}^{2}}{1-t^{2}}dt=\sum_{k=1}^{\infty }\frac{1}{2k(2k+1)}Z_{k}^{2}. \end{array}$$



The proof of the above two equalities will be provided in Appendix B. One can see the equivalence of these two equalities by using a change of variable.

Given different probabilities (*p*), Table 1 presents the critical values $c_{p}$ for which $p=P(\chi_{n}^{2}\left(\tau \right)\leq c_{p})$ for different *n*, where $\chi_{n}^{2}\left(\tau \right)=\sum_{k=1}^{n}\lambda_{k}\left(\tau \right)Z_{k}^{2}$ and calculations of critical values for finite *n* are based on Imhof’s method [16] implemented in R package CompQuadForm [21]. A few critical values are tabulated in Anderson and Darling [22] for $\chi_{n}^{2}\left(\tau \right)$ with n=∞. One can see the critical values converge very quickly as *n* increases to *∞*.

In fact, we can connect the limit distribution of WC statistic and its functional limit distribution by the Karhunen–Loève expansion of the integral of the weighted square of Brownian bridge in terms of the Jacobi polynomials. Theorem 1 immediately implies the following asymptotic distribution as n→∞.

**Corollary 1.** 
*Under the assumptions of Theorem 1, when n→∞,*

$$\frac{S_{n}\left(Y;\tau ,2\right)}{\sigma^{2}}\to_{d}\sum_{k=1}^{\infty }\lambda_{k}\left(\tau \right)Z_{k}^{2}.$$



One can check $\sum_{k=n}^{\infty }\lambda_{k}\left(\tau \right)Z_{k}^{2}\to_{p}0$ by Markov’s inequality. Hence, $\sum_{k=1}^{n-1}\lambda_{k}\left(\tau \right)Z_{k}^{2}$ converges to $\sum_{k=1}^{\infty }\lambda_{k}\left(\tau \right)Z_{k}^{2}$ in probability as n→∞. By the functional limit theorem,
$$\frac{S_{n}\left(Y;\tau ,2\right)}{\sigma^{2}}\to_{d}\left\{\begin{array}{c} \int_{0}^{1}\frac{{\left\{B\left(t\right)-tB\left(1\right)\right\}}^{2}}{1-t^{2}}dt,\;\mathrm{if}\;\tau =0,\\ \int_{0}^{1}\frac{{\left\{B\left(t\right)-tB\left(1\right)\right\}}^{2}}{t(1-t)}dt,\;\mathrm{if}\;\tau =n/2,\\ \int_{0}^{1}\frac{{\left\{B\left(t\right)-tB\left(1\right)\right\}}^{2}}{2t-t^{2}}dt,\;\mathrm{if}\;\tau =n. \end{array}\right.$$

### 2.3. Graphical Model

Assume the $\left\{Y_{i,j},1\leq i\leq n,1\leq j\leq q\right\}$ are independent and have common mean $E\left(Y_{i,j}\right)=\mu_{i}$ and variance $\mathrm{Var}\left(Y_{i,j}\right)=\sigma_{i}^{2}$. Consider testing
(7)$$H_{0}:\mu_{i}\equiv \mu ,\sigma_{i}^{2}\equiv \sigma^{2}\;vs\;H_{a}:\mu_{i}=\left\{\begin{array}{c} \mu_{-},\;\mathrm{for}\;i\leq k^{*},\\ \mu_{+},\;\mathrm{for}\;k^{*}>i, \end{array}\right.\;\mathrm{or}\;\sigma_{i}^{2}=\left\{\begin{array}{c} \sigma_{-}^{2},\;\mathrm{for}\;i\leq k^{*},\\ \sigma_{+}^{2},\;\mathrm{for}\;k^{*}>i, \end{array}\right.$$
where $\mu_{-}\neq \mu_{+}$ or $\sigma_{-}^{2}\neq \sigma_{+}^{2}$, the parameters μ, $\mu_{-}$, $\mu_{+}$, $\sigma^{2}$, $\sigma_{-}^{2}$, and $\sigma_{+}^{2}$ are unknown.

A graphical model can be established by treating each *q*-dimensional vector as a node and assigning the Euclidean distance between any two vectors. Here, we consider a path P with an ordering of nodes $\left(v_{1},\ldots ,v_{n}\right)$ and edges $\left(v_{i},v_{i+1}\right)$ for i=1,…,n−1. Associated with the path, the count of edges that connect nodes between arbitrary two disjoint sets $N_{k}=\left\{1,\ldots ,k\right\}$ and ${\bar{N}}_{k}=\left\{k+1,\ldots ,n\right\}$ is defined to be:(8)$$C_{P}\left(N_{k},{\bar{N}}_{k}\right)=\sum_{i=1}^{n-1}I\left[\left\{\left(v_{i}\in N_{k}\right)\cap \left(v_{i+1}\in {\bar{N}}_{k}\right)\}\cup \{\left(v_{i+1}\in N_{k}\right)\cap \left(v_{i}\in {\bar{N}}_{k}\right)\right\}\right],$$
where I(·) is an indicator function that takes 1 if true otherwise 0. The $C_{P}\left(N_{k},{\bar{N}}_{k}\right)$ counts edges between two groups $N_{k}$ and ${\bar{N}}_{k}$.

Denote the expectation and variance of $C_{P}\left(N_{k},{\bar{N}}_{k}\right)$ under n! permutations of nodes as $E_{\mathrm{perm}}C_{P}\left(N_{k},{\bar{N}}_{k}\right)$ and ${\mathrm{Var}}_{\mathrm{perm}}C_{P}\left(N_{k},{\bar{N}}_{k}\right)$. By [23],
$$E_{\mathrm{perm}}C_{P}\left(N_{k},{\bar{N}}_{k}\right)=\frac{2k(n-k)}{n}\;\mathrm{and}\;{\mathrm{Var}}_{\mathrm{perm}}C_{P}\left(N_{k},{\bar{N}}_{k}\right)=\frac{2k(n-k)\{2k(n-k)-n\}}{n^{3}-n^{2}}.$$

A WC statistic may be constructed as
(9)$$S_{n}\left(P;\tau ,\gamma \right)=\sum_{k=1}^{n-1}w_{k}^{-1}\left(\tau \right){\left\{-C_{P}\left(N_{k},{\bar{N}}_{k}\right)+\frac{2k(n-k)}{n}\right\}}^{\gamma }.$$

A large value of observed $S_{n}\left(P^{*};\tau ,\gamma \right)$ based on the shortest Hamiltonian path (SHP), $P^{*}$, indicates a rejection of the null hypothesis, i.e., there is a change point; see the heuristic algorithm of SHP in Biswas et al. [24] and the analysis of power and change point in Shi, Wu and Rao [25], Shi, Wu and Rao [26] for γ=2 and $w_{k}\left(\tau \right)={\mathrm{Var}}_{\mathrm{perm}}C_{P}\left(N_{k},{\bar{N}}_{k}\right)$. Here, we will establish the asymptotic distribution of $S_{n}\left(P;\tau ,\gamma \right)$ for γ=1,2. First, we give the following Lemma.

**Lemma 1.** For *$k=\left\lfloor tn\right\rfloor$ with 0<t<1,*
(10)$$\frac{1}{\sqrt{2n}}\left\{-C_{P}\left(N_{k},{\bar{N}}_{k}\right)+\frac{2k(n-k)}{n}\right\}\to_{d}{\left\{B\left(t\right)-tB\left(1\right)\right\}}^{2}-t\left(1-t\right),n\to \infty .$$

By the functional limit theorem,
(11)$$\sqrt{\frac{n}{2}}S_{n}\left(P;\tau ,1\right)\to_{d}\left\{\begin{array}{c} \int_{0}^{1}\frac{{\left\{B\left(t\right)-tB\left(1\right)\right\}}^{2}}{1-t^{2}}dt+\log\left(2\right)-1,\;\mathrm{if}\;\tau =0,\\ \int_{0}^{1}\frac{{\left\{B\left(t\right)-tB\left(1\right)\right\}}^{2}}{t(1-t)}dt-1,\;\mathrm{if}\;\tau =n/2,\\ \int_{0}^{1}\frac{{\left\{B\left(t\right)-tB\left(1\right)\right\}}^{2}}{2t-t^{2}}dt+\log\left(2\right)-1,\;\mathrm{if}\;\tau =n, \end{array}\right.$$
and
(12)$$\frac{1}{2}S_{n}\left(P;\tau ,2\right)\to_{d}\left\{\begin{array}{c} \int_{0}^{1}\frac{{\left[{\left\{B\left(t\right)-tB\left(1\right)\right\}}^{2}-t\left(1-t\right)\right]}^{2}}{1-t^{2}}dt,\;\mathrm{if}\;\tau =0,\\ \int_{0}^{1}\frac{{\left[{\left\{B\left(t\right)-tB\left(1\right)\right\}}^{2}-t\left(1-t\right)\right]}^{2}}{t(1-t)}dt,\;\mathrm{if}\;\tau =n/2,\\ \int_{0}^{1}\frac{{\left[{\left\{B\left(t\right)-tB\left(1\right)\right\}}^{2}-t\left(1-t\right)\right]}^{2}}{2t-t^{2}}dt,\;\mathrm{if}\;\tau =n, \end{array}\right.$$
which solves an open problem in [25,26]. Different values of γ lead to different rates of convergence and different ”normings”.

### 2.4. Normal Mixed Model

Assume $Y_{i,j}=\mu_{i}+U_{i}+e_{i,j},$ where 1≤i≤n, 1≤j≤q, $e_{i,j}$ are independent and identically normally distributed with mean zero and variance $\sigma^{2}$, and $U_{i}$ are independent latent variables following a normal distribution with mean zero and variance $\nu^{2}$.

Consider testing
(13)$$H_{0}:\mu_{i}\equiv \mu \;vs\;H_{a}:\mu_{i}=\left\{\begin{array}{c} \mu_{-},\;\mathrm{for}\;i\leq k^{*},\\ \mu_{+},\;\mathrm{for}\;k^{*}>i, \end{array}\right.$$
where $\mu_{-}\neq \mu_{+}$, the parameters μ, $\mu_{-}$ and $\mu_{+}$ are unknown, and we tentatively assume the time $k^{*}$, called the change point, and the variances $\sigma^{2}$ and $\nu^{2}$ to be known.

The marginal log-likelihood function of μ under $H_{0}$ is
$$\ell \left(\mu \right)=\ell_{0}-\frac{\sum_{i=1}^{n}{\left({\bar{Y}}_{i•}-\mu \right)}^{2}}{2\nu^{2}+2\sigma^{2}/q},$$
where $\ell_{0}$ does not depend on μ and ${\bar{Y}}_{i•}=\sum_{j=1}^{q}Y_{i,j}/q.$

Therefore,
$$\max_{\mu }\ell \left(\mu \right)=\ell_{0}-\sum_{i=1}^{n}\frac{{\left({\bar{Y}}_{i•}-{\hat{\mu }}_{1,n}\right)}^{2}}{2\nu^{2}+2\sigma^{2}/q},$$
where ${\hat{\mu }}_{t_{1},t_{2}}=\sum_{i=t_{1}}^{t_{2}}{\bar{Y}}_{i•}/\left(t_{2}-t_{1}+1\right)$.

In a similar way, the marginal log-likelihood function of $\mu_{-}$ and $\mu_{+}$ under $H_{a}$ can be obtained. Then, the marginal log-likelihood ratio is
$$\sum_{i=k^{*}+1}^{n}\frac{{\left({\bar{Y}}_{i•}-{\hat{\mu }}_{1,n}\right)}^{2}}{\nu^{2}+\sigma^{2}/q}-\sum_{i=1}^{k^{*}}\frac{{\left({\bar{Y}}_{i•}-{\hat{\mu }}_{1,k^{*}}\right)}^{2}}{\nu^{2}+\sigma^{2}/q}-\sum_{i=k^{*}}^{n}\frac{{\left({\bar{Y}}_{i•}-{\hat{\mu }}_{k^{*}+1,n}\right)}^{2}}{\nu^{2}+\sigma^{2}/q},$$
which is equal to
$$\frac{n{\left\{\sum_{i=1}^{k^{*}}\left({\bar{Y}}_{i•}-{\hat{\mu }}_{1,n}\right)\right\}}^{2}}{k^{*}\left(n-k^{*}\right)\left(\nu^{2}+\sigma^{2}/q\right)}.$$

As the change point $k^{*}$ could be unknown in practice, we may sum over $k^{*}=1,\ldots ,n-1$ and consider the average value, which leads to
(14)$$S_{n}\left({\bar{Y}}_{•};n/2,2\right)=\sum_{k=1}^{n-1}w_{k}^{-1}\left(n/2\right){\left\{\sum_{i=1}^{k}\left({\bar{Y}}_{i•}-{\hat{\mu }}_{1,n}\right)\right\}}^{2}.$$
where ${\bar{Y}}_{•}={\left({\bar{Y}}_{1,•},\ldots ,{\bar{Y}}_{n,•}\right)}^{⊤}$.

By Theorem 1 and Remark (3) in terms of weighted version for any τ, as n→∞,
$$\frac{S_{n}\left({\bar{Y}}_{•};\tau ,2\right)}{{\left(n-1\right)}^{-1}{\left\{\sum_{i=1}^{n}\left({\bar{Y}}_{i•}-{\hat{\mu }}_{1,n}\right)\right\}}^{2}}\to_{d}\sum_{k=1}^{\infty }\lambda_{k}\left(\tau \right)Z_{k}^{2}.$$

### 2.5. Poisson Mixed Model

Assume $Y_{i,j}$ follows a Poisson distribution with conditional mean $E\left(Y_{i,j}|U_{i}\right)=\exp\left(\rho_{i}+U_{i}\right)$. Consider testing
(15)$$H_{0}:\rho_{i}\equiv \rho \;vs\;H_{a}:\rho_{i}=\left\{\begin{array}{cc} & \rho_{-},\;\mathrm{for}\;1\leq i\leq k^{*},\\ \; & \rho_{+},\;\mathrm{for}\;k^{*}<i\leq n, \end{array}\right.$$
where $\rho_{-}\neq \rho_{+}$, the parameters ρ, $\rho_{-}$ and $\rho_{+}$ are unknown. Under normal distribution for $U_{i}$, the likelihood ratio contains an integral. With the focus on the simple Poisson mixed model without a change point, Hall et al. [27,28] applied the Gaussian variational approximation (GVA) to approximate the integral so as to avoid solving the integral. We provide a saddle point approximation here.

The marginal log-likelihood function of ρ under $H_{0}$ is
(16)$$\ell \left(\rho \right)=\ell_{1}+\sum_{i=1}^{n}\log I_{i}\left(\rho \right),$$
where $\ell_{1}$ does not depend on *r* and $I_{i}\left(\rho \right)=\int_{-\infty }^{\infty }\exp\left\{-qe^{\rho +u}+q{\bar{Y}}_{i•}\left(\rho +u\right)-\frac{u^{2}}{2\nu^{2}}\right\}du$.

The calculation of ℓ(ρ) is hindered by the lack of a closed form of the integral $I_{i}\left(\rho \right)$. Here, we apply the saddle point approximation to the integral as shown in Lemma 2.

**Lemma 2.** 
*For the integral $I\left(\rho ;a,b,\nu^{2}\right)=\int_{-\infty }^{\infty }\exp\left\{-be^{u}+au-\frac{{\left(u-\rho \right)}^{2}}{2\nu^{2}}\right\}du$,*

$$I\left(\rho ;a,b,\nu^{2}\right)\approx {\left(\frac{a}{be}\right)}^{a}\sqrt{\frac{2\pi }{a}}e^{-\frac{{\left(c-\rho \right)}^{2}}{2\nu^{2}}},$$

*where the symbol ≈ means asymptotic equivalence and the saddle point c solves $\phi^{\prime }\left(u\right)=0$ with $\phi \left(u\right)=au-be^{u}$, i.e., c=log(a/b).*


In (Equation 16), $I_{i}\left(\rho \right)=I\left(\rho ;q{\bar{Y}}_{i•},q,\nu^{2}\right)$, so Lemma 2 gives the leading term as
$$\ell \left(\rho \right)\approx \ell_{1}-\sum_{i=1}^{n}\frac{{\left(\log{\bar{Y}}_{i•}-\rho \right)}^{2}}{2\nu^{2}},$$
and the leading term approximation to $\max_{\rho }\ell \left(\rho \right)$
$$\ell_{1}-\sum_{i=1}^{n}\frac{{\left(\log{\bar{Y}}_{i•}-{\hat{\rho }}_{1,n}\right)}^{2}}{2\nu^{2}},$$
where ${\hat{\rho }}_{t_{1},t_{2}}=\sum_{i=t_{1}}^{t_{2}}\log{\bar{Y}}_{i•}/\left(t_{2}-t_{1}+1\right)$.

In a similar way, $\max_{\rho_{1},\rho_{2}}\ell \left(\rho_{1},\rho_{2}\right)$ under $H_{a}$ can be approximated, giving the approximate log-likelihood ratio
(17)$$\begin{array}{cc} \; & \sum_{i=1}^{n}\frac{{\left(\log{\bar{Y}}_{i•}-{\hat{\rho }}_{1,n}\right)}^{2}}{\nu^{2}}-\sum_{i=1}^{k^{*}}\frac{{\left(\log{\bar{Y}}_{i•}-{\hat{\rho }}_{1,k^{*}}\right)}^{2}}{\nu^{2}}-\sum_{i=k^{*}+1}^{n}\frac{{\left(\log{\bar{Y}}_{i•}-{\hat{\rho }}_{k^{*}+1,n}\right)}^{2}}{\nu^{2}}\\ \; & =\frac{n{\left\{\sum_{i=1}^{k^{*}}\left(\log{\bar{Y}}_{i•}-{\hat{\rho }}_{1,n}\right)\right\}}^{2}}{k^{*}\left(n-k^{*}\right)\nu^{2}}. \end{array}$$

Considering that the change point $k^{*}$ is unknown, we may sum (Equation 17) over $k^{*}=1,\ldots ,$n−1 as shown in (Equation 1) and consider the average value,
(18)$$S_{n}\left(\log{\bar{Y}}_{•};n/2,2\right)=\sum_{k=1}^{n-1}w_{k}^{-1}\left(n/2\right){\left\{\sum_{i=1}^{k}\left(\log{\bar{Y}}_{i•}-{\hat{\rho }}_{1,n}\right)\right\}}^{2}.$$

Note that the term $w_{k}\left(n/2\right)$ is derived from the approximate likelihood ratio statistic, different from the classical Poisson change point statistic in Csörgö and Horváth [1] (p. 27).

By Theorem 1 and Remark 3 in terms of weighted version for any τ, as n,q→∞,
$$\frac{S_{n}\left(\log{\bar{Y}}_{•};\tau ,2\right)}{{\left(n-1\right)}^{-1}{\left\{\sum_{i=1}^{n}\left(\log{\bar{Y}}_{i•}-{\hat{\rho }}_{1,n}\right)\right\}}^{2}}\to_{d}\sum_{k=1}^{\infty }\lambda_{k}\left(\tau \right)Z_{k}^{2}.$$

### 2.6. Weak Dependence

Now, we consider a space-time model for the distribution of $Y_{i,j}$, where *i* indexes time and *j* indexes space. First, we assume some weak dependence conditions on space by supposing the central limit theorem holds:(19)$$\frac{1}{\sqrt{q}}\sum_{j=1}^{q}\left(Y_{i,j}-{\bar{Y}}_{i•}\right)\to_{d}N\left(0,\sigma^{2}\right),$$
where $\sigma^{2}=\lim_{q\to \infty }q\mathrm{var}\left({\bar{Y}}_{i•}\right)$.

Next, we assume some weak dependence conditions on time by supposing that the following invariance principle or functional central limit theorem holds for any t∈(0,1) [29,30]:(20)$$\frac{1}{\sqrt{nq}}\sum_{i=1}^{\left[nt\right]}\left(Y_{i•}-{\bar{Y}}_{••}\right)\Rightarrow \tilde{\sigma }\left\{B\left(t\right)-tB\left(1\right)\right\},$$
where ${\bar{Y}}_{••}=\sum_{i=1}^{n}Y_{i•}/n={\hat{\mu }}_{1,n}$ and ${\tilde{\sigma }}^{2}=\lim_{nq\to \infty }nq\;\mathrm{var}\left({\bar{Y}}_{••}\right)$.

The weak dependence conditions in (Equation 19) and (Equation 20) are satisfied if the series is *m*-dependence, mixing, or linear process. Shao and Zhang [31] proposed a normalized change point statistic
(21)$$M_{n,q}\left(Y_{•}\right)=\max_{k}\frac{n}{w_{k}}{\left\{\sum_{i=1}^{k}\left(Y_{i•}-{\bar{Y}}_{••}\right)\right\}}^{2},$$
where $Y_{•}={\left(Y_{1,•},\ldots ,Y_{n,•}\right)}^{⊤}$ and $w_{k}=\sum_{i=1}^{k}{\left\{\sum_{j=1}^{i}Y_{j•}-\left(i/k\right)\sum_{j=1}^{i}Y_{j•}\right\}}^{2}+\sum_{i=k+1}^{n}{\left\{\sum_{j=i}^{n}Y_{j•}-\left(n-i+1\right)/\left(n-k\right)\sum_{j=k+1}^{n}Y_{j•}\right\}}^{2}$ is a random weight.

They showed that
$$\frac{M_{n,q}\left(Y_{•}\right)}{q}\to_{d}\max_{0<t<1}\frac{{\left\{B\left(t\right)-tB\left(1\right)\right\}}^{2}}{D_{1,0,t}+D_{2,t,1}},$$
where $D_{1,0,t}=\int_{0}^{t}{\left\{B\left(s\right)-\left(s/t\right)B\left(t\right)\right\}}^{2}ds$ and $D_{2,t,1}=\int_{t}^{1}[B\left(1\right)-B\left(s\right)-\left(1-s\right)/\left(1-t\right)$${\left\{B\left(1\right)-B\left(t\right)\right\}]}^{2}ds$.

Similarly, with the same $w_{k}$ as above, we propose a randomized version of WC:(22)$$S_{n,q}\left(Y_{•}\right)=\sum_{k=1}^{n-1}\frac{1}{w_{k}}{\left\{\sum_{i=1}^{k}\left(Y_{i•}-{\bar{Y}}_{••}\right)\right\}}^{2}.$$

By the functional central limit theorem, when n→∞,
$$\frac{S_{n,q}\left(Y_{•}\right)}{q}\to_{d}\int_{0}^{1}\frac{{\left\{B\left(t\right)-tB\left(1\right)\right\}}^{2}}{D_{1,0,t}+D_{2,t,1}}dt.$$

## 3. Power and Change Point Estimation

Considering the WC statistic $S_{n}\left({\bar{Y}}_{•};\tau ,2\right)$ in (Equation 14), we now consider the power of change point test based on
(23)$$\frac{S_{n}\left({\bar{Y}}_{•};\tau ,2\right)}{{\left(n-1\right)}^{-1}{\left\{\sum_{i=1}^{n}\left({\bar{Y}}_{i•}-{\hat{\mu }}_{1,n}\right)\right\}}^{2}},$$
under the alternative hypothesis in Section 2.4. We assume some weak dependence conditions in Section 2.6. We note that (Equation 23) has the same asymptotic null distribution as (Equation 4) in Theorem 1. The asymptotic distribution is shown in Theorem 2. To establish the consistency of the test, we make a further assumption that the change point index $k^{*}$ is bounded away from the endpoints.

**Theorem 3.** 
*Assume $E\left(Y_{i,j}\right)=\mu_{i}=\mu_{-}$ if $i\leq k^{*}$, $\mu_{+}$ otherwise. Under the alternative hypothesis, the change magnitude $\Delta =\mu_{+}-\mu_{-}\neq 0$. Under a weak dependence satisfying (Equation 19) and (Equation 20), $0<\tau_{1}\leq k^{*}/n\leq \tau_{2}<1$, $\tau_{1}$ and $\tau_{2}$ are two constants, $nq\Delta^{2}\to \infty$, and $n^{3/2}q^{-1/2}\left|\Delta \right|\to \infty$, $qS_{n}\left(Y_{•};\tau ,2\right)\to_{p}\infty$.*


The proof of Theorem 3 is in Appendix E. As expected, the power of the test based on (Equation 23) increases with *n*, *q*, and the size of the change in the mean.

The estimated change point is
(24)$$\begin{array}{c} \hat{k}\left(\tau \right)=\arg\max_{1\leq k<n}w_{k}^{-1/2}\left(\tau \right)\left|\sum_{i=1}^{k}\left(Y_{i•}-{\bar{Y}}_{••}\right)\right|. \end{array}$$

We refer the reader to Bai [32,33] for some early works on the asymptotic distribution of $\hat{k}\left(n/2\right)$ and [34] for a treatment on the convergence rate of $\hat{k}\left(n/2\right)$.

## 4. Simulations

The main purpose of this simulation is to assess the effect of different values for $w_{k}\left(\tau \right)$, *n*, *q*, and change magnitude on the power of our test in (Equation 23), and that of the graph-based tests [25,26,35], as both can handle high-dimensional data, and the distance of the graph can be changed to test different changes of parameters for a fair comparison. For example, if we are not sure whether the mean or variance changes, the Euclidean distance can be used to measure the distance between any two nodes in the graph:(25)$$d_{i_{1},i_{2}}={\left\{\sum_{j=1}^{q}{\left(Y_{i_{1},j}-Y_{i_{2},j}\right)}^{2}\right\}}^{1/2};$$
see Chen and Zhang [35], and Shi, Wu and Rao [25]. Another pseudo-distance can be used
(26)$$d_{i_{1},i_{2}}^{*}=\left|Y_{i_{1},•}-Y_{i_{2},•}\right|,$$
if only the change in the mean needs to be detected; see Shi, Wu and Rao [26]. We denote the maximal test of Chen and Zhang based on Euclidean distance by MST and based on the pseudo-distance by MST*. The associated algorithm is in the R package gSeg [36]. Similarly, we denote Shi, Wu, and Rao’s test (Shi, Wu and Rao [25,26]) based on Euclidean distance by SHP and based on the pseudo-distance by SHP${}^{*}$, and the associated R package can be accessed from [37].

First, we simulate $\left\{Y_{i,j},1\leq i\leq k^{*},1\leq j\leq q\right\}$ independent standard normal random variables and $\left\{Y_{i,j},k^{*}+1\leq i\leq n,1\leq j\leq q\right\}$ independent normal random variables with mean Δ and variance 1. The critical values for α=0.05 are given in Table 1 with p=1−α. We use these critical values and generate 200 simulations with sample sizes n=40,80, dimensions q=50,100, change point locations $k^{*}=n/4,n/2,3n/4$, and change magnitude Δ=0.1,0.2.

In Table 2, we show the percentage of rejections of the null hypothesis at level 0.05 for each of the change point tests. We can see that the power of the graph-based method MST${}^{*}$ or SHP${}^{*}$ is higher than that of MST and SHP, which use the pseudo-distance for detecting changes in the mean. Interestingly, the power of the graph-based method for change point detection is still not as high as that of (Equation 23). This aspect of the comparison, which we have not seen in other literature so far, is considered a new and meaningful comparison, and at least we can claim that there is room for improvement in the change point detection of the graph-based method.

Now we look at the effect of the weights on the power. This weight $w_{k}\left(n/2\right)$ yields the highest power when the change point is in the middle; however, the $w_{k}\left(n\right)$ weight yields the highest power when the change point is near the beginning of the sequence, and conversely, the $w_{k}\left(0\right)$ weight yields the highest power when the change point is near the end of the sequence. Moreover, the power increases with increasing *n*, *q*, and Δ, which agrees with Theorem 3.

Now, we introduce a mixture distribution and slightly change the way the random variables are generated. We simulate $\left\{Y_{i,j},k^{*}+1\leq i\leq n,1\leq j\leq q\right\}$ from a mixture of two normal distributions with mixture weights (0.5, 0.5) or (0.8, 0.2), means (0, 0.2) or (0, 1), and variance always being (1, 1), which corresponds to Δ=0.1 or Δ=0.2. We keep the other settings from the previous comparison. As we expected, the difference between Table 2 and Table 3 is very small.

## 5. Data Analysis

Here, we analyze the video data provided by Dr. Mathieu Lihorea, which are available from [26]. In Lihoreau, Chittka and Raine [38], the authors used artificial pollen to attract bees and an automatic monitoring camera to capture the bee’s flight path. However, this automatic monitoring feature does not fully start recording when the bee enters and stops recording when the bee leaves, in fact in this video, the recording starts before the bee enters and does not stop when the bee leaves. Since we only care about the part of the video with bees, detecting the arrival and departure of bees helps us to automatically cut the original video. Although the video contains the interference of ants, the bees are much larger compared to the ants, so it can be assumed that the presence and departure of the bees cause a change in the mean value of the pixel values of the image.

This video has a length of 49 seconds, a frame width of 352, a frame height of 288, and a frame rate of 29.97 frames per second. Shi, Wu and Rao [26] extracted the video into n=49 images according to the rate of one frame per second. From these 49 images, we can obtain that the image positions corresponding to the bee entering and leaving are 4 and 40, respectively. Moreover, we can extract this video into more images according to the rate of 2 or 5 frames per second. So, the number of images obtained, *n*, increases to 98 or 245, and at the same time, the positions of the images corresponding to the entry and exit of the bees also change with *n*. If we call the image locations where these bees appear and leave as change points, $k^{*}$, we assume that $k^{*}/n$ is constant with respect to *n* and close to 0 or 1, respectively. In Figure 3 the first row is four images located at 4 (change point), 5, 40 (change point), and 41 from extracted 49 images; the second row is four images located at 7 (change point), 8, 79 (change point), and 80 from extracted 98 images; and the third row is four images located at 19 (change point), 20, 198 (change point), and 199 from extracted 245 images. Since the images contain R, G, and B components, we use a weighted average of the R, G, and B components and same-scale transformations on the weighted average as suggested by Shi, Wu and Rao [26].

Our quadratic weight test statistics are able to detect these two change points. We compared them to the graph-based change point estimates by applying the method of SHP${}^{*}$ and MST${}^{*}$ once to the whole sequence. As shown in Table 4, all tests are significant at a level 0.05 except the quadratic weight $w_{k}\left(0\right)$ for the size of 49 returns a *p*-value 0.067; $w_{k}\left(0\right)$ and $w_{k}(n$) give the estimates of the second and first change points, respectively; $w_{k}\left(n/2\right)$ gives the same estimates of change points as $w_{k}\left(n\right)$, and both cannot give the estimates of the second change point, such as SHP${}^{*}$ and MST${}^{*}$. Thus, we recommend these two weights $w_{k}\left(0\right)$ and $w_{k}\left(n\right)$ for detecting the departure and arrival of the bee.

## 6. Discussion

This paper mainly focuses on single change point detection. However, it is possible to extend our method and apply the WC statistic to the detection of multiple change points. An approach recommended in the literature is to select data intervals where there is evidence for a single change point. Some researchers suggested penalty procedures based on either the adaptive lasso [39] or smoothly clipped absolute deviation [40,41,42]; others applied CUSUM statistics [4,43,44,45]. As long as the aforementioned intervals have been chosen, one could use tests based on WC. If the tests are rejected for some of the intervals, then the change point can be estimated by (Equation 24).

It would also be of interest, although challenging, to consider other quadratic weights, such as $w_{k}\left(n/4\right)$ and $w_{k}\left(3n/4\right)$, as these statistics may be more powerful to detect some change points that are close to the third-quarter and quarter positions of the sequence. The eigenvalues of these quadratic terms may not have recursive formulas.

## Figures and Tables

**Figure 1 entropy-24-01652-f001:**
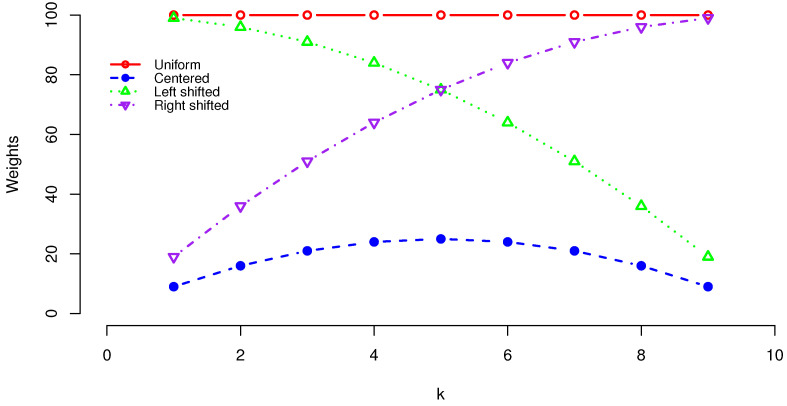
Plot of weights: $n^{2}$ (uniform), k(n−k) (centered), (n+k)(n−k) (left shifted), and k(2n−k) (right shifted).

**Figure 2 entropy-24-01652-f002:**
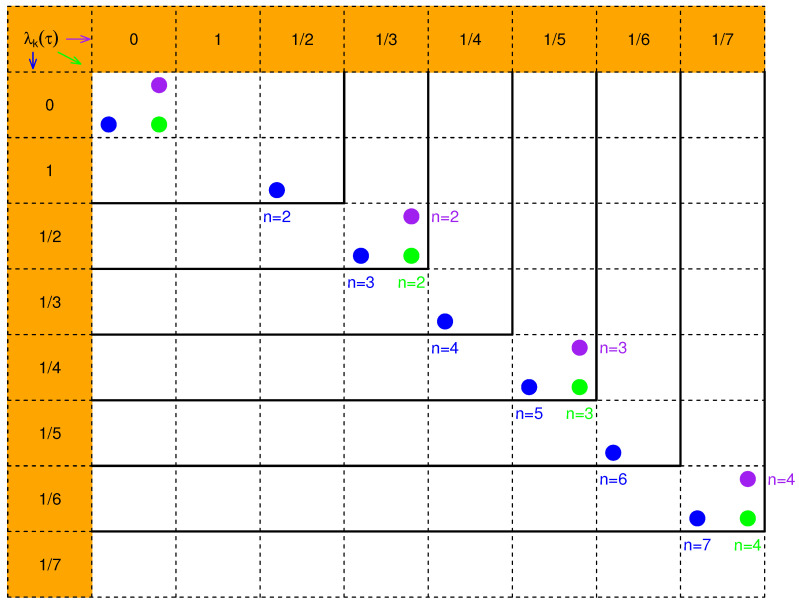
The pattern of eigenvalues (cross products of rows and columns) illustrated by dots for three weights $w_{k}\left(n/2\right)$ for τ=n/2 (blue), $w_{k}\left(0\right)$ for τ=0 (green) and $w_{k}\left(n\right)$ for τ=n (purple) with the increase of *n*.

**Figure 3 entropy-24-01652-f003:**
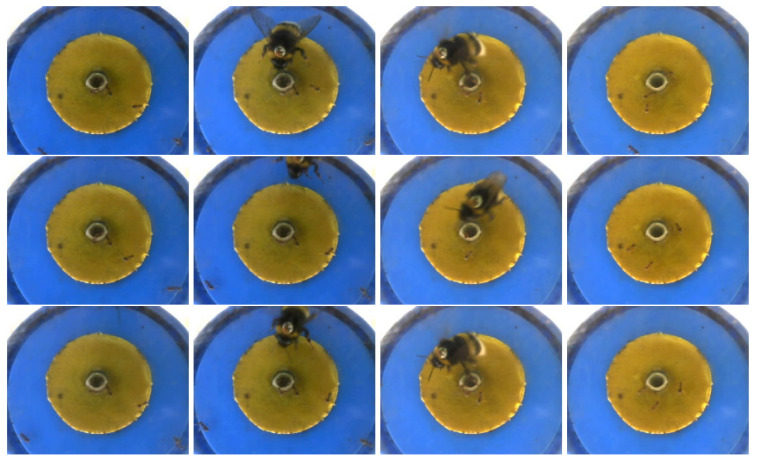
Typical images in three different image sets extracted from the same video data with different frame rates. The first row contains four images located at 4 (change point), 5, 40 (change point), and 41 from the first set of extracted 49 images (1 frame per second); the second row contains four images located at 7 (change point), 8, 79 (change point), and 80 from the second set of extracted 98 images (2 frames per second); and the third row contains four images located at 19 (change point), 20, 198 (change point), and 199 from the third set of extracted 245 images (5 frames per second).

**Table 1 entropy-24-01652-t001:** Critical values of $\sum_{k=1}^{n}\lambda_{k}\left(\tau \right)Z_{k}^{2}$ for different weights in (Equation 4), sizes (*n*), and probabilities (*p*).

		*n*									
Weight	*p*	20	40	60	80	100	200	400	1000	10,000	*∞*
$w_{k}\left(n/2\right)$	0.90	1.883	1.908	1.916	1.920	1.923	1.928	1.930	1.932	1.933	1.933
	0.925	2.111	2.136	2.145	2.149	2.151	2.156	2.159	2.160	2.161	
	0.95	2.442	2.467	2.476	2.480	2.482	2.487	2.490	2.491	2.492	2.492
	0.975	3.027	3.052	3.061	3.065	3.067	3.072	3.075	3.076	3.077	3.070
	0.99	3.828	3.853	3.861	3.866	3.868	3.873	3.876	3.877	3.878	3.850
$w_{k}\left(0\right)$	0.90	0.599	0.605	0.607	0.608	0.609	0.610	0.611	0.611	0.611	
	0.925	0.675	0.682	0.684	0.685	0.685	0.687	0.687	0.688	0.688	
	0.95	0.786	0.792	0.794	0.795	0.796	0.797	0.798	0.798	0.798	
	0.975	0.981	0.988	0.990	0.991	0.991	0.993	0.993	0.994	0.994	
	0.99	1.249	1.255	1.257	1.258	1.259	1.260	1.261	1.261	1.261	

**Table 2 entropy-24-01652-t002:** Estimated power (%) for the $w_{k}\left(0\right)$, $w_{k}\left(n/2\right)$, and $w_{k}\left(n\right)$ in (Equation 23), MST, MST${}^{*}$, SHP, and SHP${}^{*}$, based on 200 simulations; *n* are the sample sizes, *q* are the dimensions, $k^{*}$ are the change point locations, and Δ is the size of the change in the mean of the normal random variables.

*n*	40																80														
*q*	50								100								50								100						
Δ	0.1				0.2				0.1				0.2				0.1				0.2				0.1				0.2		
$\frac{k^{*}}{n}$	$\frac{1}{4}$	$\frac{1}{2}$	$\frac{3}{4}$		$\frac{1}{4}$	$\frac{1}{2}$	$\frac{3}{4}$		$\frac{1}{4}$	$\frac{1}{2}$	$\frac{3}{4}$		$\frac{1}{4}$	$\frac{1}{2}$	$\frac{3}{4}$		$\frac{1}{4}$	$\frac{1}{2}$	$\frac{3}{4}$		$\frac{1}{4}$	$\frac{1}{2}$	$\frac{3}{4}$		$\frac{1}{4}$	$\frac{1}{2}$	$\frac{3}{4}$		$\frac{1}{4}$	$\frac{1}{2}$	$\frac{3}{4}$
$w_{k}\left(0\right)$	23	41	36		67	91	84		39	72	72		96	100	100		43	74	67		97	100	100		75	96	91		100	100	100
$w_{k}\left(n/2\right)$	31	43	31		82	92	82		51	73	62		99	100	99		57	77	61		99	100	100		87	97	89		100	100	100
$w_{k}\left(n\right)$	32	43	23		89	92	64		59	73	46		99	100	91		61	74	49		100	100	97		91	97	76		100	100	100
MST	3	5	4		7	6	10		3	5	5		6	19	9		3	4	5		7	21	10		5	8	4		13	35	14
MST${}^{*}$	18	24	16		44	38	44		33	29	38		75	77	75		21	21	21		65	80	69		36	43	35		95	99	98
SHP	4	6	5		7	9	12		4	6	9		10	16	6		3	8	6		9	22	13		5	6	8		13	24	16
SHP${}^{*}$	10	13	8		33	37	33		17	23	18		67	77	65		9	12	13		49	71	54		22	32	21		90	97	92

**Table 3 entropy-24-01652-t003:** Estimated power (%) for the $w_{k}\left(0\right)$, $w_{k}\left(n/2\right)$, $w_{k}\left(n\right)$, MST, MST${}^{*}$ in (Equation 23), SHP, and SHP${}^{*}$, based on 200 simulations; *n* are the sample sizes, *q* are the dimensions, $k^{*}$ are the change point locations, and Δ is the size of the change in the mean of mixed normal distributions.

*n*	40																80														
*q*	50								100								50								100						
Δ	0.1				0.2				0.1				0.2				0.1				0.2				0.1				0.2		
$\frac{k^{*}}{n}$	$\frac{1}{4}$	$\frac{1}{2}$	$\frac{3}{4}$		$\frac{1}{4}$	$\frac{1}{2}$	$\frac{3}{4}$		$\frac{1}{4}$	$\frac{1}{2}$	$\frac{3}{4}$		$\frac{1}{4}$	$\frac{1}{2}$	$\frac{3}{4}$		$\frac{1}{4}$	$\frac{1}{2}$	$\frac{3}{4}$		$\frac{1}{4}$	$\frac{1}{2}$	$\frac{3}{4}$		$\frac{1}{4}$	$\frac{1}{2}$	$\frac{3}{4}$		$\frac{1}{4}$	$\frac{1}{2}$	$\frac{3}{4}$
$w_{k}\left(0\right)$	18	35	40		64	92	81		37	74	62		89	100	98		43	72	59		94	100	99		73	98	90		100	100	100
$w_{k}\left(n/2\right)$	28	36	37		80	94	74		50	76	57		98	100	97		53	74	52		98	100	99		84	98	87		100	100	100
$w_{k}\left(n\right)$	32	38	27		83	95	56		58	75	42		99	100	92		60	72	44		100	100	96		89	99	73		100	100	100
MST	4	5	6		4	5	4		6	6	6		5	10	6		4	4	6		4	24	20		5	6	2		8	25	20
MST${}^{*}$	20	11	24		41	44	39		27	28	27		70	72	64		22	21	24		60	75	65		38	39	33		95	100	95
SHP	4	4	8		10	15	13		6	5	10		17	28	21		8	6	5		10	29	18		9	11	6		28	54	41
SHP${}^{*}$	9	7	11		26	41	26		16	20	18		55	70	57		12	12	15		44	59	49		22	28	23		92	96	88

**Table 4 entropy-24-01652-t004:** Estimated change points for the $w_{k}\left(0\right)$, $w_{k}\left(n/2\right)$, $w_{k}\left(n\right)$, MST${}^{*}$, and SHP${}^{*}$, based on extracted 49, 98, and 245 images; *n* are the sample sizes and $k^{*}$ are the change point locations.

*n*	49			98			245	
$k^{*}$	4	40		7	79		19	198
$w_{k}\left(0\right)$		41			82			206
$w_{k}\left(n/2\right)$	4			8			19	
$w_{k}\left(n\right)$	4			8			19	
MST${}^{*}$	4			7			19	
SHP${}^{*}$	4			7			19	

## Data Availability

Not applicable.

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
