# Peer review of "A New Class of Weighted CUSUM Statistics"

_entropy, 2022, doi:10.3390/e24111652_

Round 1

Reviewer 1 Report

Please see the report.

Reviewer 2 Report

In this paper, the authors proposed a new class of weighted cusum statistics with three different types of quadratic weights accounting for different prior positions of the change points. They performed a simulation study and illustrated their application to a detection problem with video 6 data. I think the paper is well-written but some comments and suggestions should be considered before its acceptance.

1.     The word “cusum” refers to what? Is it CUmulative SUM (CUSUM)? The authors should clarify.

2.     In the abstract, replace “change-points” with “change points”.

3.     The abstract need to be rewritten with more details instead of only writing some statements.

4.     The authors should reorganize the introduction section. The authors began the introduction section with the test statistic they will use without any discussion of the problem of the study, the literature review, and the motivations of the paper.

5.     The authors should rearrange the references according to the journal format, for example, the first reference on page 1 line 16 must be [1], the second one must be [2] and so on.

6.     In equation (1), is the test statistic Sn a function of the random variables Y or the observed data y? The same thing for equation (3). I think this part needs revision.

7.     Add “and” between equations 5 and 6.

8.     The references should be updated. There is only one reference in the last four years. The authors should add some recently published papers in this regard. 
